# S100B, Actor and Biomarker of Mild Traumatic Brain Injury

**DOI:** 10.3390/ijms24076602

**Published:** 2023-04-01

**Authors:** Charlotte Oris, Samy Kahouadji, Julie Durif, Damien Bouvier, Vincent Sapin

**Affiliations:** 1Biochemistry and Molecular Genetic Department, University Hospital, F-63000 Clermont-Ferrand, France; 2Faculty of Medicine of Clermont-Ferrand, Université Clermont Auvergne, CNRS, Inserm, GReD, F-63000 Clermont-Ferrand, France

**Keywords:** S100B, biomarker, actor, traumatic brain injury, concussion

## Abstract

Mild traumatic brain injury (mTBI) accounts for approximately 80% of all TBI cases and is a growing source of morbidity and mortality worldwide. To improve the management of children and adults with mTBI, a series of candidate biomarkers have been investigated in recent years. In this context, the measurement of blood biomarkers in the acute phase after a traumatic event helps reduce unnecessary CT scans and hospitalizations. In athletes, improved management of sports-related concussions is also sought to ensure athletes’ safety. S100B protein has emerged as the most widely studied and used biomarker for clinical decision making in patients with mTBI. In addition to its use as a diagnostic biomarker, S100B plays an active role in the molecular pathogenic processes accompanying acute brain injury. This review describes S100B protein as a diagnostic tool as well as a potential therapeutic target in patients with mTBI.

## 1. Background

Traumatic brain injury (TBI), often referred to as the “silent epidemic” [1,2,3], is a growing source of morbidity and mortality worldwide. TBI is the major cause of death in people under the age of 45, and in the elderly, the incidence has recently been shown to approach epidemic proportions [4]. Each year, an estimated 69 million people worldwide sustain a TBI with a higher overall incidence in North America and Europe (1299 and 1012 cases per 100,000 people respectively [1]. The majority of sustained TBIs are reported as mild (mTBI) (81%), with moderate and severe TBIs being less prevalent (11% and 8%, respectively) [1]. In athletes, mTBI is referred to as a sport-related concussion (SRC). It is highly prevalent in contact sports such as football, hockey, rugby, soccer and basketball [5], and despite a growing interest in the recognition and management of SRCs, a large proportion still go unreported, making their identification a diagnostic challenge [6,7]. The combination of imperfect detection protocols on the field, pauci-symptomatic injuries, as well as athletes’ motivation not to report or to minimize symptoms contribute to diagnostic difficulties [8,9,10]. Altogether, there has been a significant rise in interest in diagnosing mTBI given its increasing incidence and its long-term complications, both in the general population and in the sports world. 

Cranial computed tomography (CT) is the gold standard for the evaluation of mTBI in adults and children. However, in children, several large-scale epidemiological studies have recently described that radiation exposure from CT scans increased the risk of cancer [11]. In order to reduce radiation exposure, pediatric patients with mTBI may be admitted for inpatient observation with CT scans performed only in the case of clinical deterioration. This strategy reduces radiation exposure but is approximately one-third more costly than using CT scans for initial diagnosis. In addition, most CT scans and inpatient observations could be avoided, since 93% of children suffering from mTBI have no intracerebral lesions [11]. In this context, the Pediatric Emergency Care Applied Research Network (PECARN) validated a clinical decision algorithm to help clinicians identify children with a very low risk of developing intracerebral lesions. The PECARN strategy has significantly decreased the use of CT scans, with a decrease of 90% [12].

In adults, several clinical guidelines have also been developed and proven effective in reducing the amount of negative CT scans. The most studied ones are the Canadian CT Head Rule and the New Orleans Criteria, with both originating from North America [13]. Additional guidelines, such as the National Institute for Health and Care Excellence (NICE), have been issued more recently. In 2019, Svensson et al. compared these clinical decision rules against one another in the same cohort [14]. All the reviewed guidelines allowed for an accurate identification of patients who required neurosurgical intervention and patients who died, but the most effective in reducing the amount of CT scans was the NICE guideline, recommending CT scans in 44% of patients [14]. However, among the 1353 patients included in the cohort, 825 CT scans were performed for the identification of only 70 (5.2%) cases of intracranial hemorrhage [14]. In a large French cohort of 1449 patients, 96% of scans performed were normal using the French recommendations for CT scanning [15]. 

In the field of sport, abnormalities related to concussion are typically not identified in traditional CT scanning [16]. The diagnosis of concussion requires the assessment of a combination of parameters including clinical symptoms, physical signs, cognitive impairment, neurobehavioral characteristics and sleep/wake disturbances [17]. Once asymptomatic at rest, a gradual return to activity is undertaken following the recommendations determined at the 2016 Berlin meeting [17]. However, this approach is limited by the subjectivity of clinical and neuropsychological assessments, as well as by the lack of a diagnostic gold standard [17].

Despite the use of validated guidelines in the management of mTBI in children and adults, the limiting of unnecessary scans is still insufficient and requires improvement. Improvement of the SRC management process is also encouraged in order to ensure athletes’ safety. In this setting, the use of blood biomarkers represents a relevant supplementary tool to advance clinical practice. Over the past years, a wide range of candidate biomarkers has been studied in this context [18]. These include glial (GFAP: glial fibrillary acidic protein, S100B: S100 calcium-binding protein B, MBP: myelin basic protein), neuronal (NFL: neurofilament light chain, NSE: neuron specific enolase, SBDP: spectrin breakdown products, Tau: tubulin-associated unit, UCHL1: ubiquitin carboxyl-terminal hydrolase-L1), inflammation (Interleukin 6) and immunological (autoantibodies) markers. Protein S100B is the most documented blood biomarker in this setting due to its validation in large observational and interventional studies, the routine availability of an automated assay and the precise identification of pathophysiological factors that may affect its interpretation [18]. In addition to its role as a predictive biomarker, S100B may also act as an active factor participating in pathogenic molecular processes accompanying acute brain injury [19].

## 2. What Is the S100B Protein?

### 2.1. General Characteristics

S100B belongs to the S100 family, a family of small calcium-binding cytosolic proteins first described by Moore in 1965 [20]. The name is derived from the protein’s complete solubility in a saturated ammonium sulfate solution [21]. The S100 family consists of more than twenty members characterized by two calcium-binding sites with a helix–loop–helix (“EF-hand”) structure [22]. Calcium binding induces a conformational S100 change that exposes a hydrophobic surface, allowing recruitment of other proteins leading to a biological response [23]. S100B is a small dimeric protein (molecular weight 21 kDa) that consists of ββ or αβ chains [24], predominantly expressed by astrocytes, but also to some extent by other cells in the central nervous system (CNS), including oligodendrocytes, neural progenitor cells and certain neuronal populations [25,26,27]. Physiologically, the protein has both intracellular and extracellular functions, including the regulation of protein phosphorylation and enzyme activity, calcium homeostasis and the regulation of cytoskeletal components and transcriptional factors [22]. As an extracellular factor, S100B interacts with surrounding cell types through the receptor for advanced glycation end-products (RAGE) [28]. 

### 2.2. Release and Elimination

Under physiological conditions, S100B mainly produced by astrocytes do not cross the blood–brain barrier (BBB), and the concentration of S100B in the cerebrospinal fluid is reported to be about 100-fold higher than in serum [29]. After brain insults, S100B released from damaged glial cells can diffuse into the bloodstream [30]. The mechanisms underlying this diffusion across the BBB are not completely clear. Some authors claim that S100B is released in the serum through the disrupted BBB [31,32,33]. However, in studies focused on TBI patients, there was no correlation between BBB disruption and the peak levels of S100B [26,34]. Furthermore, it has been shown that the recently described glymphatic system may play an important role in the outflow of S100B from the brain [35]. 

After diffusion into the blood stream, S100B is subject to renal elimination [26]. Some authors have reported a modest S100B elevation in patients with renal failure [36,37,38]. However, mild-to-moderate renal failure has not been shown to significantly affect S100B levels in serum [39]. S100B is eliminated with a biological half-life of approximately 30 min [39]. Diseases such as malignant melanoma or TBI may affect the half-life, with values up to 90 min [40] and 97 min [41], respectively. Note that the S100B gene is located on chromosome 21, explaining an increase in blood concentration for patients affected by Down syndrome.

### 2.3. Extracranial Sources of S100B

S100B protein is not brain specific and extracranial contributions may influence the interpretation of the results in a clinical context. In effect, S100B is also expressed in melanocytes, chondrocytes, adipocytes and skeletal muscle [42,43,44]. It is also known that serum S100B levels are influenced by skin pigmentation [45], which can be explained by a moderate production in melanocytes under physiological conditions [46]. In a cohort of 136 healthy individuals divided into three groups according to ethnicity, Black and Asian individuals had higher serum S100B concentrations than Caucasians, with mean values of 0.14, 0.11 and 0.07 µg/L, respectively [47]. Increased serum S100B levels are also observed in patients with extracranial trauma, especially in patients with bone fractures, soft tissue trauma or thoracic injury [48,49]. These damages might confound the interpretation of elevated serum S100B levels as the protein is mainly released from peripheral sources such as adipocytes, chondrocytes and skeletal muscle cells [48]. The source of S100B protein elevations is probably multifactorial and, as a recent study suggests, is associated with overall trauma severity [49]. In the context of cardiac surgery, it has been shown that increases in serum S100B levels post-surgery are not only related to cerebral hypoperfusion but also to surgical wounds, probably from surgically traumatized fat, muscle and bone marrow [50]. Moreover, numerous studies have described an increase in serum S100B levels after sports activity [51]. In fact, physical exercise and associated hypoxia are reported to induce the cerebral synthesis and release of S100B [51,52,53]. The increase in blood–brain barrier (BBB) permeability may also explain the rise in S100B during physical activity [30,51,54]. Nevertheless, the contribution of S100B from lipolysis and muscular cytolysis seems to be most plausible [51,55,56]. Due to the expression of S100B in adipocytes, authors explored the relationship between serum S100B levels and body mass index (BMI) and did not find a significant association [57]. However, it is recognized that BMI is not a direct reflection of body fat, especially in athletes, for whom this parameter leads to overestimations [51]. 

In summary, possible extracranial sources should be taken into consideration when assessing S100B levels in mTBI patients, especially in Black individuals, patients with bone fractures and athletes.

## 3. S100B as a Routine Clinical Biomarker for Management of Mild Traumatic Brain Injury

### 3.1. Routine S100B Protein Assay

S100B is a reliable biomarker, relatively unaffected by hemolysis and storage conditions, giving it appeal for use as a clinical biomarker. Serum levels remain stable for up to 8 h at room temperature and 48 h at between 2–8 °C [58]. Erythrocytes do not contain any S100B protein, thus confirming the absence of hemolytic interference in serum S100B assays, in contrast to other biomarkers such as neuron specific enolase [59]. Note that S100B is expressed in certain lymphocyte subpopulations, implying a strict adherence to pre-analytical recommendations concerning cell separation (more especially when the samples are frozen before measurement).

In routine clinical practice, the Cobas^®^ (Roche Diagnostics, Penzberg, Germany) and the Liaison XL^®^ (DiaSorin, Sangtec, Saluggia, Italy) automated immunoassays are the most frequently used systems. More recently, BioMérieux also developed an automated prototype immunoassay (Vidas^®^ 3 analyzer, bioMérieux, Marcy l’étoile, France) for serum S100B measurement without final commercialization [58]. Note that SNIBE analyzers also propose the S100B measurement. In comparison to ELISA assays, automated assays provide better analytical performance with regard to precision, linearity and accuracy, and they seem to be a preferable option for S100B determination in clinical settings [46,60]. For the two automated assays (Cobas^®^, Liaison XL^®^, DiaSorin S.p.A., Saluggia, Italy), S100B cut-off values announced by manufacturers are 0.10 and 0.15 µg/L, respectively [61]. However, the commonly accepted threshold in the management of adult patients with mTBI is 0.10 µg/L, due to the very important usage of Cobas^®^ in international publications [62]. Finally, the results differ depending on the antibodies and the type of luminescence measurement. The two automated measurements are not interchangeable, and the use of the same method is required for the monitoring of patients. The homogeneity of the results should be improved with the international standardization [46,61].

### 3.2. Addition of S100B to Guidelines in General Population

In recent years, there has been increasing interest in the identification and validation of brain biomarkers in clinical routine, and the utility of blood S100B as a brain injury marker has been documented in multiple contexts such as with circulatory arrest, stroke and TBI [63,64,65]. The protein is also associated with neurodegenerative diseases such as Alzheimer’s disease [66]. Most importantly, the protein has emerged as the most promising as a biomarker of mTBI. The potential of S100B to safely reduce CT scans was first demonstrated in a large cohort of adults with mTBI (n = 1309 patients) [67]. Since then, many observational studies have confirmed the usefulness of measuring blood S100B for the exclusion of an intracranial hemorrhage in mTBI patients [68,69,70,71]. In this context, the Scandinavian guidelines were the first mTBI clinical decision rules to include the measurement of serum S100B (Figure 1A) [72,73]. The addition of S100B measurement to the guidelines allowed a one-third reduction in unnecessary CT scans, resulting in a financial saving of approximately €39 to €71 on the cost of care per patient [74]. In a meta-analysis, Undén and Romner confirmed that low serum S100B levels (<0.10 µg/L using Cobas^®^) accurately predicted normal CT findings after mTBI in adults, provided that the sample is collected within 3 h of injury [62]. In these conditions, the sensitivity of S100B to rule out the presence of intracranial lesions was excellent (negative predictive value ~100%), with a 30% specificity [62]. Since September 2022, the French Society for Emergency Medicine (SFMU) recommends the serum measurement of S100B (sampling within 3 h post injury) for adult patients with mTBI requiring a CT scan and presenting a medium risk of complications of intracranial lesions (Figure 1B) [75]. A recent interventional study, based on 1449 patients (the largest published cohort to date), validated the inclusion of serum S100B measurement into the SFMU’s guidelines, highlighting a theoretical reduction in the number of CT scans by 32%, with a negative predictive value of 99.6% [15]. In this study, only two S100B false negatives were reported. The intracerebral lesions observed for the two patients were not progressive, meaning that they did not get worse over time and did not require neurosurgical intervention.

The French guidelines recommend a maximum delay of 3 h between trauma and blood sampling [75] rather than the 6 h suggested by the Scandinavian guidelines [72]. Because of the short half-life of S100B, the sensitivity of the biomarker could be affected by the sampling time. In their study, Laribi et al. compared S100B concentrations measured at 3 h and 6 h post-injury and found a better sensitivity with the 3 h strategy [68]. In a previous meta-analysis based on individual data from 373 children, we highlighted a sensitivity of only 90% (delay <6 h) against 97% in children whose sampling time was ≤3 h [11]. In their meta-analysis, Undén and Romner also considered that S100B should be measured within 3 h of injury [62]. Therefore, in order to avoid missing patients with intracerebral lesions in CT scans, a delay of less than 3 h would be more reliable [62,67].

To date, the evidence of the clinical utility of S100B in children is considered too low, and the biomarker is not part of the Scandinavian guidelines for the management of pediatric mTBI. In children, S100B may constitute an additional tool for the identification of low-risk patients, and it is still an area of active research.

### 3.3. S100B Specificities in the Pediatric Population

In adults, S100B concentrations did not differ according to age and sex [76], with a consensual threshold of 0.10 µg/L, although different thresholds could be proposed for patients over 65 years of age (see paragraph 3.4). Many studies reported higher S100B values in children (when compared to adults) [11,77]. In children, the biomarker’s concentrations are higher at birth and then gradually decrease during the first two years of life. A study of pediatric reference ranges using a Cobas^®^ analyzer determined in a large pediatric cohort identified three age categories with decreasing S100B levels (4–9, 10–24 and >24 months) of 0.35 μg/L, 0.23 μg/L and 0.18 μg/L, respectively [78]. More recently, Simon-Pimmel et al. provided reference ranges for infants aged 0 to 4 months, with an upper reference value of 0.51 µg/L [79]. This high value could be explained by several reasons (Figure 2). Vaginal delivery may lead to brain injury, especially in cases of prolonged labor and delivery [80], when compared to planned caesarean deliveries [81]. Another likely factor influencing S100B elevation is a difference in the permeability of the BBB and cerebral circulation. Moreover, these higher values might reflect the implication of S100B in brain maturation. These data are consistent with the neurotrophic effects of the protein at physiological concentrations, with the protein stimulating neurite outgrowth and regulating neurons survival [42]. In this sense, Bouvier et al. found a significant correlation between serum S100B concentrations and head circumference, defined using the equation [46]: S100B value (μg/L) = −1.884 × head circumference (meters) + 1.0455 (r² = 0.93).

In the literature, only a few studies interpret the S100B using age-dependent thresholds [77,82].

In our meta-analysis that included 373 children, specific reference ranges (0.35 μg/L from 0 to 9 months old, 0.23 μg/L from 10 to 24 months old, 0.18 μg/L for >24 months old) were fundamental to the interpretation of S100B levels [11]. A single threshold yielded a specificity of only 18%, while adjusting the threshold according to age-related normal values increased the specificity to 38% while maintaining 100% sensitivity [11]. This approach is consistent with recently published articles supporting the need for age-centric patient management. Recently, a French retrospective study was conducted to evaluate the impact of implementing a modified PECARN rule including the S100B protein assay for managing mTBI in children at intermediate risk for clinically important traumatic brain injury. In this study, the authors used age-dependent reference ranges for serum S100B levels as recommended by our meta-analysis. The modified PECARN rule has significantly reduced the proportion of CT scans and in-hospital observations by 34% and 45%, respectively [83]. Using age-appropriate reference values for children, the diagnostic performance of the biomarker is comparable to that observed in adults, with a reduction in the number of CT scans by approximately one-third. To confirm these promising results, a randomized, multicenter, prospective, interventional study using a stepped wedge cluster design with two arms (intervention group “S100B management” and control group “conventional management”) has been conducted in France since November 2016 (clinical trial identifier NCT02819778) [84]. The results should confirm the value of serum S100B biomonitoring in the management of mTBI in children and support the value of including S100B measurement in future guidelines (in order to reduce CT scans and hospitalizations). 

### 3.4. S100B Specificities in the Elderly Population

While children have higher levels of S100B concentrations than adults, levels of S100B in older patients following mTBI are less known. In 2013, Calcagnile et al. showed that the usefulness of S100B measurement in elderly patients may be limited by a very low specificity, reflecting a smaller decrease in the number of CT scans performed [85]. Similar results were observed in a cohort of 1449 patients including 504 patients over 65 years old [15]. In this cohort, the threshold of 0.10 µg/L resulted in a 33% reduction in CT scans performed in adults versus 19% in older patients [15]. Recently, we have confirmed that S100B levels were considerably affected by aging in a larger cohort of patients ≥65 years old suffering from mTBI with a medium risk of intracranial lesions [86]. As an adjustment of the S100B level was necessary in older patients, we proposed the use of a new 0.15 µg/L threshold for the Cobas^®^ analyzer in the routine management of patients aged between 80–90 years [86]. This threshold helped achieve a reduction in CT scans in the 80–90 years category similar to that in adult patients (~33%) [86]. In addition, for patients over 90 years old, we do not recommend the measurement of S100B. The reduction in the number of scans allowed is considerably hampered despite the use of an age-adapted threshold [86]. Several hypotheses may be formulated to explain the increase in blood S100B with aging (Figure 2). In healthy humans, an increase in BBB permeability is observed with aging, resulting in an increase in S100B blood concentrations [87]. Another hypothesis concerns an increase in β-amyloid plaques promoting S100B synthesis [88,89]. Alterations in dendrite architecture, including changes in branching complexity, reduction in branch length, along with changes in spine morphology and reduction in spine number, may also contribute to the increase in the biomarker’s concentrations [90,91]. These alterations would trigger an activation of astrocytes, resulting in a release of trophic factors such as S100B to stimulate the regrowth of dendrites [92,93,94]. In addition, neuroinflammation associated with the physiological process of aging (“inflammaging”) would also activate astrocytes cells, leading to a release of astrocytic biomarkers such as GFAP or S100B [93,94,95]. While astrocyte activation may first be neuroprotective during normal aging, experimental data from selected central nervous system pathologies suggest that if not resolved in time, reactive gliosis can exert inhibitory effects on neuroplasticity and CNS regeneration [96]. These data are consistent with the effects of S100B protein. Nanomolar concentrations of S100B exert neurotrophic effects by stimulating neurite outgrowth and regulating the survival of neurons, while micromolar concentrations are neurotoxic [97]. Further studies are needed to better understand the role of each mechanism.

### 3.5. S100B Specificities in the Athletic Population

It is known that physical exercise results in a brief increase in S100B levels through extra-cerebral synthesis and/or increased BBB permeability (Figure 2) [51]. Indeed, many studies reported higher serum S100B levels after intense exercise, such as running or swimming [98,99,100,101,102]. In one study, S100B increased simultaneously to creatine kinase and myoglobin, suggesting the potential value of S100B as a biomarker of acute muscle damage after running [98]. Although serum S100B concentration has been shown to rise in relation to exercise alone, the S100B increase has been described as higher in response to the number of contacts in many sports such as hockey, American football or rugby [103,104,105,106]. Indeed, in these sports, athletes are exposed to impacts, often repetitive to the head and that do not necessarily cause signs of concussion [107]. In competitive elite soccer, serum concentrations of S100B were found to be significantly correlated to the number of headers [108]. In American football, it has been shown that blood S100B levels were elevated in post-game measures compared with the respective pre-game values. An increase in the frequency and magnitude of head impacts, without a concussion being detected, resulted in the largest acute changes in S100B plasma levels [107,109,110]. In professional rugby players, a significant increase in S100B concentration was reported within 2 h following a game (without concussion), and this increase was correlated with the number of body collisions during a match [104,111]. Since the authors did not observe a significant correlation between S100B and creatine kinase, the increase in S100B would most likely be related to sub-concussive head impacts [109,111].

These data are essential for proper interpretation of S100B values when suspecting a concussion. This is particularly important as the implementation of a blood biomarker in the SRC screening process may enhance the diagnostic performances and improve the overall management of concussed players. When screening for SRCs, we recommend interpreting S100B levels through a personalized medical approach, by determining for each player a baseline measured away from any sport practice [112]. This approach reduces the influence of inter-individual variability subsequent to concussion history [112]. We also recommend assessing the intra-individual variability to establish the effect of contact sport on the player’s baseline over a season. In these conditions, we have recently demonstrated the interest of incorporating S100B measurement to the Head Injury Assessment (HIA) protocol in the management of concussed professional rugby players [112]. The HIA protocol combines cognitive, balance, and memory evaluation, and consists of three assessments: immediately post-injury (HIA-1), within 3 h of the injury (HIA-2), and a follow-up at 36–48 h post-injury (HIA-3) [113]. A concussion will be considered unresolved if the HIA-3 examination is still positive [113]. In our study, an individual increase in S100B blood concentration (related to baseline) measured at 36 h post-match was predictive of non-resolutive concussion [112]. To conclude, the introduction of personalized measurement of serum S100B in the HIA-3 clinical assessment may help in a more objective identification of non-resolutive concussions, allowing a better protection for the players, in line with the recommendations of the 2016 Berlin meeting [17]. Further studies are expected to confirm the value of other biomarkers in the management of SRCs.

### 3.6. Anti-S100B Antibodies, a Complementary Biomarker?

Functional BBB changes following TBI cause potential S100B protein to enter the peripheral bloodstream as «foreigner», leading to the initiation of an autoimmune response and the development of S100B autoantibodies [114,115,116]. In addition to S100B, anti-S100B autoantibodies may serve as blood-based biomarkers for brain injury, although the evidence is sparse [114]. A study conducted in children reported high levels of anti-S100B autoantibodies in the first days after severe TBI indicating failure of compensatory-adaptive immunological mechanisms and high permeability of the BBB, which are poor prognostic signs in this context [117]. Autoimmunity triggered following GFAP release into the bloodstream has also been reported in human TBI. A study including TBI patients showed an average 3.77-fold increase in anti-GFAP autoantibody levels by day 7–10 post-injury. This increase in autoantibody levels was negatively correlated with outcome at 6 months [118]. In another study, elevated anti-S100B antibodies have been observed in football players with repeated sub-concussive episodes characterized by BBB disruption. Serum levels of S100B auto-antibodies also predicted persistence of diffusion tensor imaging scan abnormalities which in turn correlated with cognitive changes [119]. In the context of TBI, the measurement of anti-S100B along with S100B represents a promising approach but requires further investigations to establish the diagnostic value of combining both in a model for concussion screening. Note that the presence of macro-analytes (circulating conjugates of analytes with immunoglobulins) is a well-known source of interference in immunoassays. Macro-analytes are high molecular weight conjugates that are measurable through the available immunoassays despite being biologically inactive. The potential analytical interference of S100B/IgG autoantibody complex, referred to as “macro-S100B”, in the S100B assay should be investigated by polyethylene glycol (PEG) precipitation and gel filtration chromatography [120]. In principle, polyethylene glycol precipitation may be applied to any immunoassay in which a macro-complex interference is suspected, this process has been extensively described in the assessment of macroprolactinemia [121]. Overall, this interference would result in an overestimation of S100B levels and may lead to a decrease in the specificity of the biomarker when screening for intracranial lesions in the management of mTBI patients. In professional athletes, who are prone to these autoantibodies, “macro-S100B” may lead to the mismanagement of the players.

## 4. S100B as an Actor of Mild Traumatic Brain Injury 

In addition to being a biomarker reflecting intracranial lesions, S100B, via an interaction with the receptor for advanced glycation end-products (RAGE), is an important mediator involved in damage consecutive to TBI. Indeed, depending on its local concentration, and the density of RAGE molecules expressed on the surface of sensitive cells, S100B can exert trophic or toxic effects. At nanomolar concentrations and in the presence of relatively low RAGE cell density in the extracellular environment, S100B exerts protective and neurotrophic effects [122], resulting in a stimulation of neurite outgrowth, the promotion of neuronal survival [97,123] and the prevention of motor neuron degeneration in newborn rats after sciatic nerve section [124]. These effects suggest an important role of S100B in astrocyte–neuron communication during brain development, as well as a protective effect on neurons during the initial phases of brain injuries [125]. Conversely, at submicromolar–micromolar concentrations in the presence of relatively high RAGE cell density, the protein participates in the brain inflammatory response [122]. The ligation of S100B to RAGE activates the nuclear factor kappa B (NF-kB) signaling pathway, which in a positive feedback loop will increase the expression of RAGE [126]. This RAGE/NF-kB axis activation increases the expression of proinflammatory cytokines [127] and plays an important role in cellular processes such as neurodegeneration, excitotoxicity, oxidative stress and micro-oedema [128].

In a context favoring chronically elevated extracellular concentrations, such as TBI, S100B may also behave as a Damage-Associated Molecular Pattern (DAMP) molecule, or alarmin. Recently, Zou et al. indicated that the activation of the S100B/RAGE signaling pathway may be an important factor in mediating the damage of TBI [129]. In their study, the level of RAGE expression in brain tissue and astrocytes, and the level of soluble RAGE (sRAGE) in serum, were both upregulated by TBI [129]. Physiologically, sRAGE is hypothesized to counteract the detrimental action of RAGE as a competitive inhibitor of the signaling pathway and as a ligand scavenger [126,130]. Recently, we hypothesized that concussion resolution at 36 h in professional rugby players may be related to an elevation in sRAGE, which may regulate blood levels of S100B and thus preserve brain function [112]. Furthermore, the S100B/RAGE pathway is also involved in chronic traumatic encephalopathy (CTE), a complication that affects some athletes participating in contact sports and exposed to repetitive head impacts [131]. In this context, increased extracellular S100B binds to neuronal RAGE receptors leading to hyperphosphorylation of Tau and contributing to neurofibrillary tau tangle formation [132]. Causal relationships between sub-concussive impact-induced S100B elevation, neuronal tau aggregation and later-onset of CTE require longitudinal investigations [109,133,134]. 

## 5. S100B as a Putative Therapeutic Target 

In both animal and cellular models, TBI triggers the elevation of S100B in the brain tissue and serum [129]. The release of S100B, by interacting with RAGE through paracrine and systemic responses, results in additional synthesis and secretion of the S100B by astrocytes [129]. After TBI, the activation of the S100B/RAGE signal is known to aggravate brain tissue injury [129]. In experimental TBI, the inhibition of S100B using a S100B knockout mice model resulted in functional and neuropathological improvement compared to non-genetically modified mice [135]. In this context, targeting S100B and its receptor RAGE could be beneficial for the treatment of neurological disorders such as TBI [136]. The targets for potential therapeutic use are summarized in Table 1. Arundic acid is a novel drug acting through the inhibition of S100B synthesis in astrocytes [137]. In animal and astrocyte TBI models, the administration of the S100B inhibitor arundic acid led to significant attenuation of RAGE expression and serum sRAGE levels [129]. Moreover, the administration of a RAGE antagonist reversed the S100B increase, and subsequently attenuated RAGE expression and sRAGE secretion [129]. The inhibition of RAGE also attenuated TBI-induced brain and lung damage and improved astrocyte viability after stretch injury [129]. While exogenous administration of sRAGE has never been evaluated in a TBI model, it has been shown to reduce the development and progression of Alzheimer’s disease in animal models [138]. Another treatment in mice with a neutralizing S100B antibody has been found to reduce TBI-induced lesion volume, attenuate microglial activation, improve neuronal survival, induce improvement in retention memory function and reduce sensorimotor deficits [135]. In addition, the S100B inhibitor polydatin has been shown to have a therapeutic effect in severe TBI, both on TBI-induced neuronal damage and secondary lung injury. By inhibiting S100B expression, this molecule promotes lung vascular permeability recovery and attenuates the oxidative stress response and inflammatory cytokines release [139]. Papaverine may be another potential therapeutic target for acute TBI, potentially via the RAGE-NF-κB signaling pathway and by possibly inhibiting microglia activation. In a recent experimental TBI model in mice, single-dose papaverine treatment reduced brain edema, neuroinflammation and apoptosis. Moreover, the administration of papaverine provided microglial inhibition with a decrease in serum S100B levels and showed neuroprotective effects [140]. Nevertheless, it also has been reported in another study that intraventricular administration of S100B may improve cognitive functions and increase hippocampal synaptogenesis in TBI animals [141]. This discrepancy could be, as previously mentioned, explained by the duality of S100B actions related to its concentrations [142]. Indeed, while S100B acts as a neurotrophic factor at low concentrations, an overproduction by activated glia can lead to exacerbation of neuroinflammation and neuronal dysfunction [142].

In summary, results from experimental animal model studies indicate that the protein may be regarded as a therapeutic target in TBI. To date, clinical trials on the use of arundic acid or sRAGE have only been conducted in patients with neurodegenerative diseases. Further studies are required in TBI patients to gain a more in-depth understanding of the S100B/RAGE signaling modulation and to determine the therapeutic dose for a pharmaceutical manipulation of RAGE [143]. The targeting of cerebral cells by drugs interacting with the S100/RAGE pathway is a pharmacological challenge. This is a major issue in the management of contact sport athletes, as repetitive concussive and subconcussive impacts can lead to CTE. While waiting for a targeted therapeutic management, nutritional supplementation represents an attractive approach to reduce the deleterious effects of SRCs [144]. It is known that cerebral nutritional needs are modified during sport practice, especially in contact sports [144]. Concussion causes massive consumption of cellular ATP and produces a state of anaerobic hyper-glycolysis which results in the accumulation of lactate, calcium sequestration and also aerobic, which results in mitochondrial dysfunction, leading to altered oxidative metabolism [145]. In this sense, antioxidants may represent an interesting therapeutic option, especially as RAGE interacts with S100B to generate reactive oxygen species (ROS) involved in oxidative stress and long-term neurodegeneration [138]. Tissue reconstruction is also accelerated after brain injury, generating increased need of essential nutrients including some omega-3 polyunsaturated fatty acids and amino acids [144].

## 6. Concluding Remarks 

S100B protein has emerged as the most used biomarker for clinical decision making in the management of patients with mTBI. It is the most studied and documented blood biomarker in this setting, and its application has been validated in numerous observational and interventional studies. Another advantage of S100B over other potential biomarkers lies in the availability of an automated assay that is suitable for emergency practice. In adults, S100B measurement has been incorporated into Scandinavian and French guidelines for reducing the number of CT scans performed and hospitalizations in the context of mTBI. With a threshold of 0.10 µg/L (using Cobas^®^) and 0.15µg/L (for Liaison XL^®^), allowing for a sensitivity of virtually 100%, S100B measurement upon admission results in a significant reduction in CT scans (~30%). Overall, while S100B has high potential as a diagnostic biomarker, its limitations, particularly its lack of specificity and time dependent sensitivity, suggest that it may need to be used in conjunction with other diagnostic tools and clinical assessment. Moreover, the threshold level for detecting mTBI in adults is not applicable to all populations. In children and people over 80 years old, specific cut-offs must be determined to address the biomarker’s lack of specificity. In competitive sports, we recommend an interpretation of S100B levels based on a baseline level specific to each athlete, allowing for a personalized monitoring in the case of concussion. The baseline measurement should be performed at rest, avoiding an increase in the biomarker’s concentrations related to sports practice. In addition, interference related to the presence of anti-S100B autoantibodies should be investigated to assess the potential impact on the interpretation of results. Finally, in addition to its use as a diagnostic biomarker, S100B protein constitutes a promising therapeutic target in cerebral lesions since it plays an active role in the molecular pathogenic processes accompanying acute brain injury. In this context, more research is needed to determine the safety and efficacy of targeting S100B in human patients with TBI.

## Figures and Tables

**Figure 1 ijms-24-06602-f001:**
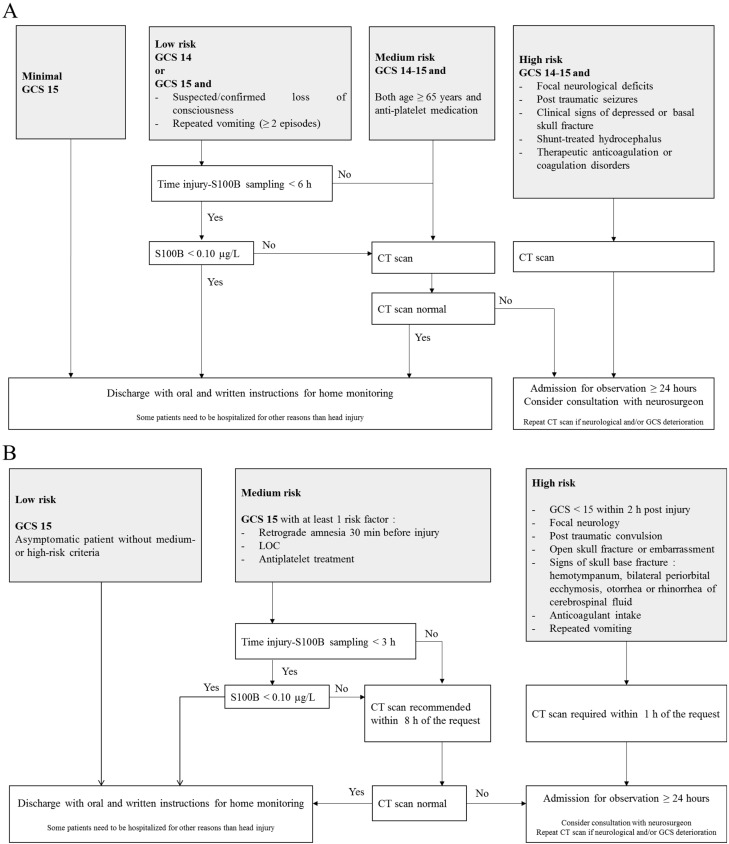
Scandinavian (**A**) and French guidelines (**B**) for the management of patients with mild traumatic brain injury. CT scan: Computed tomography scan; GCS: Glasgow Coma Scale; LOC: Loss of Consciousness.

**Figure 2 ijms-24-06602-f002:**
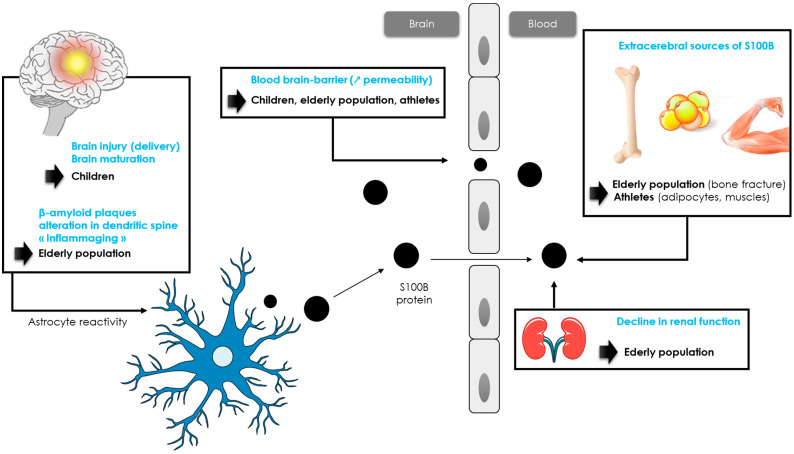
Hypotheses of S100B protein increase in children, athletes and elderly population.

**Table 1 ijms-24-06602-t001:** S100B and RAGE as putative therapeutic targets.

	Molecule	Mechanism	Effects (TBI Models)	Reference
S100B	Arundic acid	Inhibition of S100B expression in astrocyte	↘ RAGE + sRAGE expression in brain	Zou et al., 2022 [129]
	Polydatin		↘ neuronal damage ↘ secondary lung injury ↗ lung vascular permeability recovery ↘ oxidative stress, inflammatory cytokines	Gu et al., 2021 [139]
	Anti-S100B	Neutralizing S100B antibody	↘ TBI-induced lesion volume ↘ microglial activation ↘ sensorimotor deficits ↗ neuronal survival ↗ retention memory function	Kabadi et al., 2015 [135]
RAGE	RAGE antagonist FPS-ZM1	Blood–brain barrier permeant blocker of RAGE V domain-mediated ligand binding	↘ S100B ↘ RAGE + sRAGE expression in brain ↘ TBI-induced brain and lung damage ↗ astrocyte viability after stretch injury	Zou et al., 2022 [129]
	sRAGE	Competitive inhibitor of the RAGE pathway and ligand scavenger	? (never evaluated in a TBI model do date)	Prasad et al., 2019 [138]
	Papaverine	Inhibition of RAGE/NF-κB signaling pathway	↘ brain edema ↘ neuroinflammation ↘ apoptosis ↘ S100B ↗ neuroprotective effects	Saglam et al., 2021 [140]

RAGE: receptor for advanced glycation End-products; sRAGE: soluble RAGE; TBI: traumatic brain injury.

## Data Availability

Not applicable.

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
