# Peer review of "S100B, Actor and Biomarker of Mild Traumatic Brain Injury"

_ijms, 2023, doi:10.3390/ijms24076602_

Round 1
Reviewer 1 Report
In the manuscript presented by Oris et al., the authors reviews the literature on S100B as a biomarker, actor and therapeutic target for mild traumatic brain injury. They described the necessity of determine a blood biomarker for mTBI to reduce the amount of negative CT scans. The authors also discussed the evidence supporting S100B as a reliable biomarker and its high diagnostic sensitivity. In addition, S100B as an actor and a putative therapeutic target were summarized. The structure and contents are well organized. The manuscript is very well written and easy to be understood. The conclusion is solid and provides us with clear future directions on solutions for ensuring youth with brain injury be properly identified, kept from being misidentified, and effectively supported in schools. In addition, the language is fluent and precise. However, there are a few issues that need to be addressed before publication.
1. The limitations or challenges associated with using S100B as a biomarker and therapeutic target for mTBI need to be discussed.
2. The number of the sub-heading “S100B as a putative therapeutic target” needs to be changed from 4.2 to 5, which will be consistent with the title “S100B: biomarker, actor and therapeutic target of mild traumatic brain injury”. Otherwise, the title need to be modified.
Author Response
In the manuscript presented by Oris et al., the authors reviews the literature on S100B as a biomarker, actor and therapeutic target for mild traumatic brain injury. They described the necessity of determine a blood biomarker for mTBI to reduce the amount of negative CT scans. The authors also discussed the evidence supporting S100B as a reliable biomarker and its high diagnostic sensitivity. In addition, S100B as an actor and a putative therapeutic target were summarized. The structure and contents are well organized. The manuscript is very well written and easy to be understood. The conclusion is solid and provides us with clear future directions on solutions for ensuring youth with brain injury be properly identified, kept from being misidentified, and effectively supported in schools. In addition, the language is fluent and precise. However, there are a few issues that need to be addressed before publication.
We thank the reviewer #1 for the positive appreciation of our manuscript.
- The limitations or challenges associated with using S100B as a biomarker and therapeutic target for mTBI need to be discussed.
We added the following sentences (page 17):
- “Overall, while S100B has high potential as a diagnostic biomarker, its limitations, particularly its lack of specificity and time dependent sensitivity, suggest that it may need to be used in conjunction with other diagnostic tools and clinical assessment. Moreover, the threshold level for detecting mTBI in adults is not applicable to all populations”
- “In this context, more research is needed to determine the safety and efficacy of targeting S100B in human patients with TBI”.
- The number of the sub-heading “S100B as a putative therapeutic target” needs to be changed from 4.2 to 5, which will be consistent with the title “S100B: biomarker, actor and therapeutic target of mild traumatic brain injury”. Otherwise, the title need to be modified.
The number of the sub-heading “S100B as a putative therapeutic target” was changed from 4.2 to 5 (page 14).
Reviewer 2 Report
In this well written review paper, the authors describe the role of S100B as a biomarker in the management of mild TBI. They further discuss its physiological actions at nanomolar and micromolar concentrations, and provide a brief overview of molecules that target S100B or its receptor (RAGE).
· Fix the typo in the first line of abstract. It mentions “mTBI accounts for 80% of all mTBI cases”, it should be 80% of all TBI cases.
· Page 3, second paragraph. Provide reference for the sentence “Protein S100B is the most documented blood biomarker in this setting due to its validation in large observational and interventional studies, the routine availability of an automated assay, and the precise identification of pathophysiological factors that may affect its interpretation.”
· Page 11, section 3.6. Fix typo “Autoimmunity trigged following GFAP”, to triggered.
· It should also be mentioned that even in the Scandinavian guidelines S100B biomarker is not used for initial management of minor head trauma in children as the evidence was considered too low and the number of studies too few.
· Although it is mentioned that S100B measurement in adults at a threshold of 0.10ug/L has “virtually” 100% sensitivity. The authors should address the legal liability of missed cases.
· Table 1 is not mentioned in the manuscript. Add a sentence in the text (page 14), describing the contents of table 1.
Author Response
In this well written review paper, the authors describe the role of S100B as a biomarker in the management of mild TBI. They further discuss its physiological actions at nanomolar and micromolar concentrations, and provide a brief overview of molecules that target S100B or its receptor (RAGE).
We thank the reviewer #2 for the positive appreciation of our manuscript
- Fix the typo in the first line of abstract. It mentions “mTBI accounts for 80% of all mTBI cases”, it should be 80% of all TBI cases.
The typo was fixed in the first line of abstract (page 2).
- Page 3, second paragraph. Provide reference for the sentence “Protein S100B is the most documented blood biomarker in this setting due to its validation in large observational and interventional studies, the routine availability of an automated assay, and the precise identification of pathophysiological factors that may affect its interpretation.”
A reference (Bouvier D, Oris C, Brailova M, Durif J, Sapin V. Interest of blood biomarkers to predict lesions in medical imaging in the context of mild traumatic brain injury. Clin Biochem. 2020;85:5-11) was provided for the sentence “Protein S100B is the most documented blood biomarker in this setting due to its validation in large observational and interventional studies, the routine availability of an automated assay, and the precise identification of pathophysiological factors that may affect its interpretation” (page 4).
- Page 11, section 3.6. Fix typo “Autoimmunity trigged following GFAP”, to triggered.
The typo was fixed (page 12).
- It should also be mentioned that even in the Scandinavian guidelines S100B biomarker is not used for initial management of minor head trauma in children as the evidence was considered too low and the number of studies too few.
We mentioned the following sentence: “To date, the evidence of the clinical utility of S100B in children is considered too low, and the biomarker is not part of the Scandinavian guidelines for the management of pediatric mTBI. In children, S100B may constitute an additional tool for the identification of low-risk patients, and is still an area of active research.” (page 8 and 9).
- Although it is mentioned that S100B measurement in adults at a threshold of 0.10ug/L has “virtually” 100% sensitivity. The authors should address the legal liability of missed cases.
We added the following sentence: “In this study, only two S100B false negatives were reported. The intracerebral lesions observed for the two patients were not progressive, meaning that they did not get worse over time, and did not require neurosurgical intervention.” (page 8).
- Table 1 is not mentioned in the manuscript. Add a sentence in the text (page 14), describing the contents of table 1.
We added the following sentence: “The targets for potential therapeutic use were summarized in Table 1” (page 14).